# The Effects and Costs of Personalized Budgets for People with Disabilities: A Systematic Review

**DOI:** 10.3390/ijerph192316225

**Published:** 2022-12-04

**Authors:** Marguerite Robinson, Marie Blaise, Germain Weber, Marc Suhrcke

**Affiliations:** 1Luxembourg Institute of Socio-Economic Research (LISER), 4366 Esch-sur-Alzette, Luxembourg; 2Faculty of Psychology, University of Vienna, 1010 Vienna, Austria; 3Centre for Health Economics, University of York, York YO10 5DD, UK

**Keywords:** personal budgets, disability, quality of life, cost-effectiveness

## Abstract

This article reviews the peer-reviewed and grey literature published from January 1985 to November 2022 that has quantitatively evaluated the effects of personalized budgets for people with disabilities (PwDs), in terms of a range of benefit and cost outcomes. Benefit metrics of interest comprised measures of well-being, service satisfaction and use, quality of life, health, and unmet needs. A search was conducted using the PsycINFO, MEDLINE, CINAHL, ASSIA, and Social Care Online databases. Based on inclusion criteria and a quality assessment using the Downs and Black Checklist, a final count of 23 studies were identified for in-depth review. Given the heterogeneous nature of the studies, a narrative synthesis, rather than a formal meta-analysis, was undertaken. Taking the relatively scarce and often methodologically limited evidence base at face value, the findings suggest that—overall—personalized budget users tend to benefit in terms of well-being and service satisfaction outcomes, with the exception of mixed effects for people with mental health conditions. Only a minority of studies have investigated the cost-effectiveness or costs-only of personalized budgets, finding mixed results. Two out of the three cost-effectiveness studies find personal budgets to be more cost-effective than alternative options, meaning that the possibly higher costs of personalized budgets may be more than outweighed by additional benefits. Some evidence looking at service use and/or costs only also points to significant reductions in certain service use areas, which at least hints at the potential that personalized budgeting may—in some cases—entail reduced costs. Further research is needed to explore the generalizability of these conclusions and to better capture and understand the factors driving the observed heterogeneity in some of the results.

## 1. Introduction

The United Nation Convention on the Rights of Persons with Disabilities (UNCRPD) has been ratified by 82 countries, entering into force in 2008 [1]. It was the fastest convention ever negotiated—only taking four years—with the highest number of signatories on its first day, reflecting the universal support for the ideas expressed in the Convention. Based on nine general principles, including—among others—respect, dignity, and inclusion in society, it provides a complete typology of disabilities as well as a description of all the rights of People with Disabilities (PwDs), to ensure that they can fully assert their rights regardless of disability types. The treaty is intended as a social tool to change attitudes and mentalities towards PwDs by promoting their rights and ability to make decisions on their own, with a focus on the concepts of ‘consent’ and ‘autonomy’. Following on from the Independent Living Movement (ILM), emerging in the 1960s in the United States (US), the UNCRPD also set out the right of independent living, with the goal of enabling PwDs to have a choice over where and how they want to live [2]. 

Following this direction, in high-income countries, service delivery has been making progressive shifts towards a person-centred approach to disability services in place of the institutional or agency-based care provision that previously characterized the care landscape [3]. To this end, the implementation of consumer-directed personal assistance care and the provision of personal budgets has progressed. Many different labels are commonly used to refer to the fundamental concept underlying consumer-directed personal assistance, most frequently ‘self-direction’, ‘self-determination’, ‘personalization’, and ‘individualized care’. While those concepts tend to differ slightly in their processes and implementation, they all share the inherent goal of empowering PwDs by imparting greater levels of choice and control over their care services, in particular, on their choice of personal assistant(s) [4]. However, it has been noted that freedom of choice alone is not necessarily tantamount to improved outcomes, if the abilities of service users to self-manage their health and social care is not also enhanced [5]. It is also uncertain what outcome metrics should be used to measure improvements (or lack thereof), as there exist various options to measure service user outcomes in social care, including satisfaction, quality of life, health, social inclusion, unmet needs, etc. It is nonetheless important to empirically assess whether more personalized models of assistance—as desirable and worthy as their aspiration surely is—do, in practice, live up to their expectations, in terms of the observed improvement in relevant outcomes.

Much of the research conducted to date has focused on qualitative outcomes for service users, as it is often difficult to quantify individual perceptions of the subjective concepts of satisfaction and quality of life [6]. While the quantitative evidence appears to be more limited, a number of studies at both the national and subnational level have been conducted. The limited evidence base for quantitative evaluations has been highlighted by previous reviews in this area; in particular, the low number of cost-effectiveness evaluations and a clear need for larger-scale high quality studies has been identified [7]. Harkes et al. [8] conducted a systematic review of self-directed support for people with learning disabilities and, while evidence of improved quality of life was reported, limited access to services and consideration of service users’ range of abilities were identified as limitations to care provision. However, this review was limited to the United Kingdom (UK) and included only two quantitative studies. Similar conclusions were reached in a systematic review by Webber et al. [7], who focused on the effectiveness of personal budgets for people with mental health conditions, and by Fleming et al. [9], who covered the literature until 2016, reviewing seven quantitative studies from the US and UK. The authors examined the effects and experiences of individualized budgeting, limiting their analysis to adults with disabilities.

In the present rapid systematic review, carried out in accordance with the PRISMA statement [10], we add to previous work by also taking into account the most recent quantitative evidence that has attempted to evaluate outcomes, costs, and cost-effectiveness of personalized assistance or budgeting and by covering not only the peer-reviewed but also the grey literature, as well as non-English literature (i.e., French and German). In light of the differences in outcomes and the specificities of the interventions and evaluation methods, we focused on a narrative review rather than a formal meta-analysis. In some contrast to previous reviews, we also take an inclusive approach, covering both adults and children with disabilities and their caregivers. We also present and discuss our findings by disability type, to detect potentially relevant heterogeneities. For instance, people with intellectual disabilities might be expected to face greater difficulties in handling personalized budgets compared to people with physical disabilities, thus requiring external support. Intellectual disability is often associated with psychiatric disabilities, referring to dual-diagnosis of specific mental health disorders, including psychosis, personality disorders, major depression, and bi-polar disorders. These are characterized by fluctuating sequels, where periods of well-adapted functioning are followed by periods of dysfunctional behaviours. This might interfere in the capacity to handle personalized budgets, especially if there is no appropriate support during these periods. Physical disability refers to limitations in motor functioning, which includes not only mobility issues, but also apraxia and loss of motor speech. Assuming no dual-diagnosis, these people have a higher capacity of comprehension and, therefore, higher levels of self-determination, facilitating access to personalized budgets and assistance.

## 2. Methods

### 2.1. Search Terms and Data Sources

A preliminary search of existing reviews and relevant grey literature was conducted to identify keywords to inform the final search terms. The Population, Intervention, Comparison, Outcomes, and Study (PICOS) framework was used to structure these keywords and to further define the search strategy (see Appendix A).

The population included three categories of disability: physical disabilities, intellectual and developmental disabilities, and mental health conditions. A physical disability is a condition affecting one’s mobility, physical capacity, and endurance, that may occur either at birth, due to genetic problems, or following an accident or a disease. Intellectual disability includes disorders of cognitive–intellectual development characterized by etiological heterogeneity, such as Down Syndrome, a condition caused by a chromosomal aberration (trisomy 21). Developmental disability refers to a disability starting in the developmental age (i.e., before individuals reach 18 years of age) and that may be associated either with physical and/or intellectual disabilities. On the contrary, mental health disabilities are based on a psychiatric diagnosis. According to the International Classification of Diseases (ICD) [11], mental health disorders encompass, for instance, depression, anxiety disorders, post-traumatic stress disorder, psychosis, schizophrenia, bi-polar disorders, and personality disorders.

Interventions included studies where consumers had control of their care decisions, with different models or forms of personalized budgeting. It is important to acknowledge that, in the existing literature, there does not yet appear to be a clear consensus on the use and definition of key terms around ‘personalized budgeting’. This may be due to work in this area being recent and consumer-directed care systems, differing considerably in their implementation and processes. With this in mind, for the purpose of our study, a generic, very broad definition of ‘personalized budgeting’ was chosen. We define a personalized budget as an amount of money allocated to or for a person with a disability to pay for their care and support needs. This includes all forms of self-directed or consumer-directed care, as long as service users are in control of their care, being able to choose and direct their services and providers.

Comparisons were made between the traditional system that would allocate the budget to an institution and different forms of personalized budgeting.

For this review, we defined seven outcome categories to capture a broad set of potential consequences of personalized budgets of primary interest:Health and physical well-being,Quality of life and psychological well-being,Unmet needs,Satisfaction with services,Service use,Costs, andCost-effectiveness.

It is important to note that a variety of scales and measurement tools have been used in the literature to determine outcomes and benefits for service users. We provide details on the specific metric and tool used in the detailed summary of our reviewed studies. To account for potential heterogeneity between the studies, outcomes from the studies are presented by main type of disability group (incl. ‘mixed disabilities’), highlighting the evidence for each group and any variation in the preferences and outcomes for PwDs. Outcomes are presented and discussed separately for service users on one hand and, on the other hand, for parents or other caregivers that are ultimately involved in the decision-making process and are also impacted by the quality of care services.

The search framework comprised studies including quantitative components. Mixed-methods studies were also included in the search strategy, provided that relevant quantitative results could be extracted. Where possible, the Cochrane rapid review recommendations [12] were followed. According to the Cochrane definition, a rapid review “is a form of knowledge synthesis that accelerates the process of conducting a traditional systematic review through streamlining or omitting specific methods to produce evidence for stakeholders in a resource-efficient manner” [12] (p.1).

The following databases were searched: PsycINFO, MEDLINE, CINAHL, ASSIA, and Social Care Online. Databases were searched in the Title and Abstract fields only. Preliminary scoping searches indicated that using the full PICOS strategy defined above was too restrictive without searching the full-text documents. Thus, only the Population and Intervention keywords from the PICOS framework were used, as the predefined Outcomes and Study Design keywords severely limited the search results and would have resulted in missing relevant documents.

### 2.2. Inclusion and Exclusion Criteria

The search strategy included both the academic and grey literature from the start of 1985 to mid-November 2022. To be included in this review, studies had to meet the criteria detailed below (see Table 1). They were eligible where there was a comparison with an alternative model of care, via either a control group or a pre-post study design. Those studies exclusively addressing the care of older people were excluded, as the disability and old-age sectors often differ in their goal settings as well as in their financing strategies. Articles including elderly participants were deemed eligible whenever the share of elderly covered did not represent most of the total study population.

### 2.3. Quality Assessment

Methodological quality and risk of bias were assessed for each article using the Downs and Black Checklist [13]. The 27-item tool assesses quality in 5 domains: reporting, external validity, internal validity, confounding/selection bias, and statistical power. This tool was chosen as it is applicable to both randomized and non-randomized studies and proved more efficient in the context of a rapid review. For the purpose of this review, the quality assessment has been carried out at the article level, in light of observed notable differences between the articles that covered the same intervention, in terms of, e.g., research question, outcomes reported, the implementation process across sites, and the targeted population.

## 3. Results

### 3.1. Search and Selection Results

The search of electronic databases yielded 3351 potentially relevant articles from the start of 1985 to August 2020. Database-specific search strategies and search results are detailed in Appendix A). An additional 88 articles were identified through expert knowledge and screening of the reference lists of existing reviews and other relevant articles. Once duplicates were removed, a final count of 2181 articles was screened at the level of title and abstract by 1 reviewer (MR). A consistency check was conducted by a second reviewer (MS) who screened titles and abstracts of a random selection of 370 (17%) excluded articles. Conflicts arose with seven articles and were resolved through discussion. In all, 288 articles were identified for full text screening, however, the full-text for 16 articles could not be located and a further 13 articles were identified, at later stage, from the reference lists of articles going forward to full-text screening. One reviewer (MR) screened the 285 full-text articles deemed eligible for inclusion. Uncertainties were discussed with a second reviewer (MS), and any conflicts were resolved in discussion among three reviewers (MR, MS, GW). A list of excluded articles and a brief description is provided in Appendix A.

The record search was updated to cover the period from August 2020 to mid-November 2022, using the same search strings (i.e., the keywords defined through the PICOS framework were used). Medline, Psycinfo, CINAHL, Social Care online, and ASSIA were searched (see Appendix A, for updated figures). In total, 980 records were found. After removing duplicates, 532 records were screened at the title level and then at the abstract level. Fourteen articles were analysed in more detail, but most were qualitative evaluations. The process is outlined in Figure 1.

A final count of 33 articles, spread across 20 intervention programmes, was found to meet the inclusion criteria and assessed by a single reviewer (MR) following the Downs and Black checklist. The outcome of the assessment was discussed with a second reviewer (MS), who re-screened studies scoring less than 50% for verification of the result. The full quality assessment is detailed in Appendix A. A commonly adopted criterion for grading article quality assigns a quality grade based on the total score, as follows: excellent (24–28), good (19–23), fair (14–18), or poor (<14) [14]. Of the 33 articles assessed, 1 was of excellent quality, 10 were deemed to be of good quality, 12 were classified as fair, and 10 were classified as poor quality.

The *Reporting* domain of the Downs and Black checklist assesses whether information provided in each article is sufficient to allow for an unbiased assessment of the findings. The maximum score obtainable is 11, and the average score of the 23 included articles was 7.74 with a range of 6–10. The main weaknesses in this domain were a lack of clearly defined confounders and the reporting of individuals that were lost to follow-up. All articles scored full marks for providing a comprehensive statement of their aim and clearly describing participant characteristics, main outcomes to be measured, intervention of interest, and main findings.

The representativeness of the article findings and their generalizability to the target population is assessed in the *External Validity* domain. A maximum score of 3 is possible, and the average scored was 1.56 with a range of 0–3. Five articles received the maximum score, with most others failing to report response rates.

Biases in the measurement of the intervention and outcomes are addressed in the *Internal Validity* domain of the checklist, with a maximum possible score of 7. For the implementation of personal budgets, blinding of the participants is not possible, and all articles were graded 0 on this criterion. The average score (marked out of 7) was 4.61, with a range of 4–5. No article focused on interventions that attempted to blind those measuring the outcomes. Another limitation was the lack of adjustment for different lengths of follow-up of the participants, to account for cases where individuals were in receipt of the treatment for different lengths of time. All articles reported using validated and reliable outcome measures.

The *Confounding* domain measures bias in the selection of participants. Nine articles received the maximum score of 6, and the average score was 4.22, with a range of 2–6. The weakest point was the lack of concealing intervention assignments from both patients and health care staff until the recruitment was complete. Furthermore, only 10 of the 23 articles randomly assigned individuals to treatment or control groups. The final domain of *Statistical Power* determines if a sufficient sample size to detect a behaviourally important effect is considered. Only seven articles conducted such an analysis.

The main areas of weakness of the ten articles rated as *poor quality* were in the domains of external validity and cofounding, with average scores of, respectively, 33.3% and 30%. For the purposes of this review, these articles were judged to be of insufficient methodological quality to contribute to the evidence base. The remaining 23 articles were deemed to be of acceptable methodological quality for inclusion in this review. The scores for each article and each domain are shown in Figure 2 (for more details, see Appendix A).

### 3.2. Article Characteristics

A single reviewer extracted data from the 23 articles included in this review using a pre-agreed-upon extraction table, Appendix A. Of those, 21 covered service user outcomes, while 2 reported on caregiver outcomes. All articles were published between 1998 and 2022. Sixteen (70%) focused on the US, four on England and the UK, and three on Italy.

Mental health conditions were the sole focus of eight articles. Two considered outcomes for people with physical disabilities. Another four focused on people with intellectual disabilities and developmental disabilities, although one of these noted the high incidence of major secondary diagnoses amongst the targeted population, including physical disabilities and mental health conditions. Nine articles presented outcomes covering a diverse range of disabilities. Sample sizes varied significantly between articles, with the smallest covering a participant population of just 45 people and the largest 2825. Article characteristics are summarized in Table 2 and in more detail in Appendix A.

A diverse range of study designs was observed amongst the included articles. Ten were RCTs (or Randomized Controlled Trials), two of which reported on caregiver outcomes, and one was a non-randomized controlled trial, which adopted a ‘pragmatic design’ with some degree of randomization. Six undertook cross-sectional surveys with a control group, and a further four adopted a before–after controlled comparison. Two did not use a control group and opted for a pre-post comparative design.

Most articles focused on working age adults, with two presenting outcomes for children aged 3–17 [24,28]. Where possible, elderly participants were removed from the results presented in this review. However, many articles only reported mean age and not upper age limit, and it is not possible to provide an accurate estimate of the proportion of elderly individuals in the evidence base. Three articles explicitly defined the older populations [16,26,27]). However, it was not possible to remove these from the results, as outcomes were presented for the whole sample (see Table 3).

The terminology used to describe the personal assistance and personalized budgeting services varied amongst the studies, as did the implementation models. Four articles focused solely on consumer-directed personal assistance services, in which service users could hire and manage their caregivers. The remaining nineteen included some form of monetary benefit allocated to the service user, although they all differed on fundamental characteristics, including eligibility, budget size, who could be hired, and what service or goods could be purchased.

The largest intervention programme, the US Cash and Counseling demonstration, gave consumers a monthly allowance to hire workers and purchase care-related services and goods. The funding model allowed for a flexible use of the allowance, including hiring family members, and provided counselling and fiscal support to service users. Other models implemented some form of individual or individualized budget as a means for service users to self-direct their personal care services. A particularly unique model was that presented in Croft et al. [22], in which mental health service users could choose to intentionally reduce their service use and apply the resulting cost savings to fund the purchase of approved non-clinical goods and services to aid in achieving their recovery goals.

Two large intervention programmes in England, the Individual Budgets Pilot Programme [29,37] and the Personal Health Budget Pilot Programme [27], differed from each other in their funding streams. The former piloted a budget model, which brought together multiple funding streams, including social care, housing-related support, and equipment budgets, so that an individual would be eligible for a single sum that could be spent flexibly following their priorities and preferences. In contrast, the personal health budget programme was funded solely by the Department of Health and allows budget holders to purchase a wide range of services and support, including social care, well-being, and therapy related services.

To account for potential heterogeneity, results are presented by disability group, highlighting the evidence for each group and any variation in the preferences and outcomes for PwDs. Outcomes are also presented separately for children and for parents or other caregivers that are ultimately involved in the decision-making process and are also impacted by the quality of care services.

### 3.3. Outcomes for Services Users

A large variety of outcome measures are used in the literature to quantify cost and user benefits from personalized budget models. In this section, outcomes are presented by disability sub-group where possible. Studies covering a mix of disabilities are presented separately and outcomes specific to a disability type are indicated where applicable. Outcomes found to be statistically significant are summarized in Table 4, and further details on measurement scales and outcomes can be found in Appendix A.

#### 3.3.1. Physical Disabilities

Two articles focused solely on individuals with physical disabilities [15,30]. Both followed a cross-sectional survey design and examined consumer-directed care, which allowed users to hire and manage their own personal assistants. Satisfaction with services was reported by both, and Hagglund et al. [30] further reported on quality of life and unmet needs.

Satisfaction with Services:

Beatty et al. [15] measured satisfaction using a 16-item Personal Assistance Satisfaction Index. Participants receiving consumer-directed care had significantly higher total satisfaction, compared with those receiving non-consumer-directed services. Consumer-directed participants were significantly more likely to be *extremely* or *very satisfied* with the costs of care, their choice and control over care, their control over the assistant’s work schedule, their authority to direct their assistant, and the availability of their assistant in both everyday care and emergency situations. The control group, not being eligible for consumer-directed care, were paying for services out-of-pocket. To remove the impact of this from the satisfaction outcome, a further analysis was undertaken by re-computing the satisfaction score after subtracting the item for cost, and the difference remained significant.

Hagglund et al. [30] used two instruments to measure user satisfaction: the Patient Satisfaction Questionnaire (PSQ-III) and the Group Health Association of America (GHAA). Three satisfaction factors were derived: service quality, daily living satisfaction, and community living satisfaction. Compared with agency-directed services, consumer-directed participants reported greater satisfaction in all three factors, with significantly higher ratings for daily living satisfaction and community living. Satisfaction with service quality was also higher amongst consumer-directed participants, but the difference was not significant.

Quality of Life:

Hagglund et al. [30] reported quality of life measured from the SF-36 under the two criteria of emotional and social well-being and physical well-being. Compared with agency-directed services, consumer-directed participants reported a greater improvement in quality of life in both criteria, but the difference was not significant.

Unmet Needs:

Unmet needs were measured by Hagglund et al. [30] using an adapted version of the Client Questionnaire, which defined unmet service needs as the number of times in the past month when an individual was not able to do ADLs or IADLs because help was not available. Both agency-directed and consumer-directed participants reported high levels of unmet needs, including unmet bowel and bladder needs and being able to eat when hungry. While consumer-directed participants reported fewer unmet needs, there were no statistically significant differences between the two groups.

#### 3.3.2. Intellectual and Developmental Disabilities

Four articles focused on adults with intellectual and developmental disabilities [18,20,24,28]), although Conroy et al. [20] includes many individuals reporting major secondary disabilities including mental illness and physical disabilities. The first two refer to interventions that allocated an individual or individualized budget to participants, included a control group, and performed a before–after comparison. Caldwell et al. [18] used a longitudinal approach and reported outcomes for the treatment group at three time points (baseline, four-year mid-point, and nine-year follow-up). In addition, a treatment–control comparison was performed at the nine-year point; however, the control group was only recruited to the programme at this time. The Conroy et al. [20] evaluation was conducted across three distinct pilot sites, with a fourth site serving as a comparison group, although it was only a strict comparator with one pilot site. The two others [24,28] were RCTs focusing on the Cash and Counseling demonstration in the US and reported separately on outcomes for children in Florida, aged 3–17 years, with developmental disabilities. Caldwell et al. [18] reported on the outcomes of service satisfaction and unmet service needs from the family perspective, and Conroy et al. [20] presented several quality-of-life outcomes. Foster et al. [28] reported both service satisfaction and unmet need outcomes for children in Florida, and Dale et al. [24] reported on annual expenditures and costs per month.

Satisfaction with Services:

Caldwell et al. [18] used a five-item questionnaire to measure service satisfaction, in which families indicated to what degree they received the service they needed. At the nine-year time point, families in the program were significantly more satisfied with services compared to families on the waiting list for consumer-directed care. Furthermore, over time, between baseline and follow-up, consumer-directed care led to a significant increase in service satisfaction. Satisfaction for children in Florida was measured on four-point scales and covered satisfaction with the caregiver’s schedule, the relationship with the caregiver, help around the house and community, and overall care arrangements [28]. Compared with the control group, children receiving a monthly allowance reported being significantly more satisfied across all four domains.

Unmet needs:

Caldwell et al. [18] used a modified version of the Family Support Index to gauge families’ unmet service needs for 28 common types of services. Families were asked if they used a service, and if they were not using it, they were asked if they needed it. At the 9-year time-point, families in the consumer directed program had significantly fewer unmet needs in 15 of the 28 services areas, compared with families on the waiting list. Over time, from baseline to follow-up, consumer-directed care led to significantly decreased needs in five service areas: occupational therapy, social/recreational activities, educational/vocational training, assistance obtaining benefits, and assistance obtaining vocational services. Foster et al. [28] measured unmet needs for help in the four areas of daily living activities, household activities, transportation, and routine health care. Compared with services as usual, significantly fewer children receiving the monthly allowance reported having unmet needs in all four areas.

Quality of life:

Three areas in the domain of quality of life were measured by Conroy et al. [20]: choice making (power), perception of quality of life, and adaptive and challenging behaviours. The authors’ Decision Control Inventory was used to measure the extent to which life decisions were made by paid staff versus the user and/or friends and relatives, from which a score for individual control was derived. Participants in all three sites experienced a significant increase in the power held by themselves or their allies, while the control group did not. Individuals’ perception of quality of life was measured using the Quality-of-Life Changes Scale, which covers 14 dimensions of quality, including health, friendships, and safety. All three sites, and the control group, show significant increases in their perception of quality of life. Adaptive and challenging behaviours were measured on a 14-item scale recording various maladaptive behaviours. Participants in only one treatment group site showed a statistically significant increase in adaptive behaviour, and there were no significant changes in challenging behaviours, although all groups improved slightly.

For children in Florida [28], parents of both treatment and control group members were asked how they satisfied were with the way their child was spending life. Overall, parents of children from the treatment group were more likely to report higher satisfaction with their child’s life.

Cost:

For children in Florida participating in the Cash and Counseling Demonstration, parents of the treatment group members had the opportunity to receive a monthly allowance to hire caregivers of their choice or to buy other services and goods to meet their child’s care needs [24]. The control group children continued to receive traditional waiver services. In the first year of programme enrolment, the waiver expenditures for the treatment group were significantly higher than in the control group. However, the difference between the groups in total annual average Medicaid expenditures was small and not significant (see Appendix A). The observed difference was primarily due to significantly lower expenditures on Medicaid home health services by the treatment group. In the second year of enrolment, the difference in waiver expenditures was even larger, as was the difference in total Medicaid expenditures, and both were significant.

Costs per recipient per month were also reported for children in Florida, and it was found that costs were significantly higher for the treatment group ($1378) compared with the control group ($1099).

Overall, the studies show that consumer-directed care by means of individual(ized) budgets led to greater satisfaction, fewer unmet needs, and some improvements in quality of life for people with intellectual and developmental disabilities. In some cases, such improvement in outcomes may have been associated with an increase in costs, as illustrated by the study on the children in Florida that took part in the Cash and Counseling Demonstration, whose costs and expenditures were found to be significantly higher.

#### 3.3.3. Mental Health Conditions

Eight articles reported exclusively on people with mental health conditions [21,22,23,26,27,31,32,33], adopting different study designs. The intervention programme reported by Forder et al. [27] was a large multi-site evaluation of Personal Health Budgets and included a variety of chronic health conditions. For the purposes of this review, the outcomes relevant for the mental health cohort were extracted. All articles focused on interventions involving some form of monetary payment and were conducted in the US, England, and Italy. Two reported only on costs or service use and did not report any service user benefits [22,23]. The remaining six all reported some measure of quality of life and two reported on service satisfaction.

Quality of life:

Six articles reported on quality-of-life outcome measures, all using different scales. Cook et al. (2019) reported on the outcomes of perceived level of recovery, changes in psychosocial status, and reduction in psychiatric and somatic symptoms. Perceived level of recovery was measured by the Recovery Assessment Scale and, compared with services as usual, self-directed participants improved significantly over time in Recovery Assessment Scale total scores and on two of the sub-scales of goal orientation and personal confidence. Psychosocial status was measured in the three areas of self-esteem, coping mastery (control), and the extent to which participants felt they were being served in an autonomy-supportive environment. In all three areas, the treatment group improved significantly over time when compared with the control group. Finally, reduction in psychiatric and somatic symptoms was measured with the Brief Symptom Inventory Global Severity Index. No significant difference was found between the control and treatment groups in global severity, however, the treatment group had significantly lower somatic symptoms severity over time.

Fontecedro et al. [26] compared outcomes for recipients of individual health budgets with those receiving care as usual, using the Italian version of the Health of the Nation Outcome Scale (HoNOS), which uses 12 items to detect both clinical and psychosocial problems. The total mean scale HoNOS score did not differ significantly between the two groups. However, the treatment group was found to be significantly more likely than the control group to have severe to very severe cognitive problems, be at a higher risk for severe to very severe problems related to hallucinations and delusions, and to be at a higher risk for moderately severe problems in the activities of everyday life. In contrast, treatment group members were significantly less likely to be at risk for problems in the availability of resources for work and recreation activities, compared to the control group.

Both Leuci et al. [31] and Pelizza et al. [32] focus on the use of PHB in Italy. They reported three outcome categories to measure quality of life using the same scales, i.e., Brief Psychiatric Rating Scale (BPRS), Global Assessment of Functioning (GAF), and Health of the Nation Outcome Scale (HoNOS). Pelizza et al. [32] showed that personal budget multiaxial intervention is associated with a specific improvement in negative symptoms (i.e., blunted affect, emotional withdrawal, and motor retardation) and in social problems (HoNOS) of patients with Severe Mental Illness (SMI) after the 24 months of follow-up. Leuci et al. [31] evaluated the applicability of a PHB intervention and its impact on the user quality of life in a sample of adults with First-Episode Psychosis (FEP). The PHB group reported significant improvements in functioning, in psychiatric symptoms and in behavioural problems, including aggressive behaviours and self-injury, compared with the control group. The authors suggested that psychopharmacological treatment alone is not always sufficient for people with FEP and supported the implementation of PHB as a part of an early psychosis-specific program.

Forder et al. [27] reported four outcomes in the domain of quality of life. The Adult Social Care Outcomes Toolkit (ASCOT) was used to measure care-related quality of life, which measures people’s achievements in everyday activities, including basic capabilities such as dressing and feeding and more complex capabilities such as feeling safe and having a sense of control. The personal health budget group reported higher care-related quality of life on the ASCOT scale, but the difference relative to the control group was not significant. Health-related quality of life was measured using the EQ-5D scale, in which participants rate their health status, how it has changed, and their difficulty carrying out a range of tasks. People in the personal health budget group reported a lower health-related quality of life compared to the control group, but again, the difference was not significant. Psychological well-being was measured using the General Health Questionnaire (GHQ-12) and, while the personal health budget group scored lower than the control group, no significant difference was found between the groups. Finally, subjective well-being was measured using a subjective global measure used by the Office of National Statistics in the UK’s Integrated Household Survey. The personal health budget group reported improvements in subjective well-being compared with the control group, but the change was not significant. It should be noted, however, that the small sample size in the mental health sub-group results in wider confidence intervals, and thus lower levels of statistical significance.

Focusing on Cash and Counseling, Shen et al. [33] reported on one outcome related to quality of life using a dichotomous measure of whether care recipients were very satisfied (or not) with the way life is being spent. The treatment group was found to be significantly more likely than the control group to have a better self-reported quality of life.

Service satisfaction:

Two articles reported on service satisfaction [21,33]. Client satisfaction was measured using the Client Satisfaction Questionnaire at both the 12- and 24-month follow-up points. At both time points, self-directed care participants had significantly higher satisfaction with their mental health services [21]. Shen et al. [33], using secondary data from the Cash and Counseling Demonstration, extracted outcomes for the participants with mental health conditions who had made Medicaid claims related to mental illness in the year before enrolment. Compared with the control group, consumers with a history of mental illness in the treatment group had significantly higher likelihoods of reporting to be satisfied with their caregiver’s schedule, help around the house and community, and overall care arrangements.

Service use:

Two articles reported data on service use. Croft et al. [22] reported the pre–post-program difference in the percentage of people with service use in the categories of crisis and inpatient, mental health clinical outpatient, mental health community support and coordination, and alcohol and other drug outpatient and community-based services. The percentage of individuals using mental health clinical outpatient and alcohol and other drug outpatient and community-based services decreased; however, the change was not significant. No significant change was observed in the other two service categories either. Croft et al. [23] compared service use differences between self-direction program participants and non-participants in the four service utilization categories of rehabilitation hours, outpatient treatment hours, residential days, and emergency room hours. Self-direction program participants were found to use, on average, 62.58 more rehabilitation services hours and 22.40 more outpatient treatment hours than non-participants, and the differences were significant. Self-direction participants used fewer emergency service hours and more residential days than non-participants, but the differences were not significant.

Costs:

Three articles reported costs data and provided a comparison between either the treatment and control group costs or pre–post-program costs. Over the two-year program duration, Cook et al. [21] reported lower total mean costs for the treatment group compared with the control group in the two individual years and also in both years combined ($5240 ± $5500 versus $5493 ± $8268). In terms of individual service costs, the treatment group spent less on average per person, compared with the control group, on skills training, psychosocial rehabilitation, case management, inpatient hospitalization, psychiatric crisis services, substance abuse treatment, medication management, and medications. In contrast, they spent more on average on psychotherapy, peer services, and diagnostic services. Overall, being in the treatment group reduced total service costs, but the difference was not significant.

Croft et al. [22] reported the pre–post-program difference in standardized monthly costs for the four service categories of (1) crisis and inpatient, (2) mental health clinical outpatient, (3) mental health community support and coordination, and (4) alcohol and other drug outpatient and community-based services. It was found that individuals used significantly fewer mental health clinical outpatient services after program participation compared to before, leading to significantly lower mean standardized monthly costs for this service ($38.45 versus $80.28). Standardized monthly costs were shown to decrease in all other service categories, but none were significant.

Forder et al. [27] reported differences in costs for the mental health cohort. Total costs were defined as the sum of direct and indirect costs. The personal health budget group significantly reduced their expenditure on indirect costs compared with the control group (change = −£3050). However, they reported a relative increase in expenditure on direct costs (change = £180), but the change was not significant. The reduction in total costs was larger for the personal health budget group compared with the control group (change =-−£2880) but this difference was not significant. A cost-effectiveness analysis was performed using both the ASCOT and EQ-5D outcome scales. Using ASCOT, the average net benefit was £4880 greater for people in the treatment group compared to people in the control group (*p* = 0.096), and personal health budgets were found to be cost-effective. Using the EQ-5D scale, the net benefit was also greater, £1810, but the difference was not significant.

There is evidence of significant benefits in quality of life associated with individualized budgets for mental health users, although several risks have also been recorded. Over time, costs can be reduced, and under certain conditions, individualized budgets can prove more cost-effective than alternative approaches. Service users have shown increased levels of satisfaction with services, but there is evidence to suggest that service use can increase for certain service types.

#### 3.3.4. Mixed Disabilities

Seven articles included populations with a mixed range of disabilities [16,17,19,25,29,34,35]. Five of these studies were conducted in the US and two in England and in the UK. Four were RCTs and the other three were controlled cross-sectional surveys. Four focused on consumer-directed care, allowing service users to hire and manage caregivers of their choice, while in the other three, some form of monetary payment was allocated to the service user. Six studies reported on satisfaction with services, four reported unmet needs, and a further three reported quality-of-life outcomes. Two studies reported on participant’s health perception and four conducted a cost comparison between the treatment and control groups, with two of these performing a cost-effectiveness analysis.

The evaluation of individual budgets in the UK reported outcomes for people in three disability subgroups: mental health conditions, physical disabilities, and learning disabilities [29]. Older people were also part of this trial, forming a fourth cohort. Outcomes were generally reported for the whole sample (including older people) and for each disability cohort. Outcomes specific to the older people cohort are not reported here, but the total sample outcomes presented below include older people. It should be noted that sample sizes for the individual disability cohorts were much smaller than the total sample size, especially for the mental health and the learning disability cohorts, and the evaluation authors cautioned against drawing any firm conclusions. Details of sample sizes and outcomes are reported in Appendix A and are summarized in the text below. More recently, Woolham and Benton [35] reported on the results of a cross-sectional survey of personal budget holders and traditional service users in a single English local authority and compared outcomes between people aged 65 years and older and those under 65 years with either learning disabilities, mental health conditions, or physical disabilities. Only the outcomes for the ≤64-year group are presented here.

Two sites from the Cash and Counseling demonstration, conducted across three US states, included adults with physical disabilities and who may also have had cognitive disabilities. The third site, in Florida, included adults with both physical and developmental disabilities [17,25]. The article by Benjamin et al. [16] did not specify the nature of disabilities amongst its participants, but people with severe cognitive impairment were excluded from the sampling frame. Finally, Wiener et al. [34] included adults with physical disabilities as well as individuals with intellectual and developmental disabilities.

Quality of life:

Glendinning et al. [29] reported three quality-of-life outcomes: perceived quality of life, psychological well-being, and social care outcomes (ASCOT). Program participants were asked to rate their perceived quality of life on a seven-point scale from ‘*so good it could not be better’* to ‘*so bad it could not be worse’*. For mental health service users, this self-reported quality of life was significantly higher for those in the treatment group than those in the control group. However, this difference did cease to be significant when proxy responses were removed. Amongst the other disability subgroups (physical and learning disabilities), self-reported quality of life was lower for the Individual Budget (IB) group than the control group, but the difference was not significant. Psychological well-being was measured with the 12-item General Health Questionnaire. For the total sample, there was no difference in outcomes for the treatment and control groups. The learning disability and mental health groups reported better well-being, but the difference was not significant. In contrast, the learning disability group reported poor well-being, but the difference was not significant. The ASCOT toolkit was used to measure needs for help across the seven domains of control, safety, personal care, accommodation, food and nutrition, and social participation ad occupation. Better outcomes were reported by the treatment group for the total sample and the physical activity and mental health subgroups, but these differences were not significant. Similar outcomes were reported by the treatment and control groups of the learning disability cohort. Comparing individual domains within ASCOT between the treatment and control groups at the total sample level, it was found that the treatment group was significantly more likely to report feeling in control of their daily lives compared with the control group.

In the survey conducted by Woolham and Benton [35], psychological well-being was measured using the General Health Questionnaire. The treatment group reported significantly better psychological well-being compared with the traditional services users.

Users’ quality of life was also reported by Carlson et al. [19]. The outcome is constructed based on two statements, including how satisfied users were with their way of spending life these days and whether health problems or lack of assistance limit their social activities, their educational pursuit, or their ability to work. Differences at baseline between the treatment group and the control group were not significant. Due to a restricted number of answers to quality-of-life related questions, that were asked only to non-proxy participants. This outcome was finally excluded from regression models.

Health and physical well-being:

The difference between the treatment and control groups’ self-perceived health was reported in Glendinning et al. [29], which was measured on a five-point scale from ‘*very bad’* to ‘*very good’*. For the learning disability cohort, the treatment group reported worse self-perceived health than the control group. In contrast, the mental health cohort reported better outcomes than the treatment group. However, none of these differences were significant. No difference was found between the treatment and control groups for the physical disability cohort.

Woolham and Benton [35] used the Activities of Daily Living Scale to assess program participants’ ability to carry out everyday activities. For the younger cohort, aged ≤ 64 years, the personal budget holders scored slightly higher than the control group, but there was no significant difference between the scores.

Satisfaction with services:

Service satisfaction for users receiving consumer-directed or agency-directed care was reported by Benjamin et al. [16]. Satisfaction was measured across five domains: technical quality, service impact, general satisfaction, interpersonal manner, and provider shortcomings. In the first four of these domains, consumer-directed participants rated their services significantly more positively than agency-directed participants. For the fifth domain of provider shortcomings, both groups rated care similarly.

In the Cash and Counseling RCT conducted amongst adults in three US states, Brown et al. [17] measured satisfaction across six domains: caregiver’s schedule, relationship with caregiver, help with daily living activities, help around the house/community, overall care arrangements, and help with routine health care. Across all three states, the proportion of adults reporting that they were satisfied with the first five domains of services was significantly higher for the treatment group compared with the control group. For the domain of ‘*help with routine health care’*, all adults in the treatment groups across all three states reported being significantly more satisfied than the control group; however, the difference was only significant in Arkansas and Florida. Carlson et al. [19] reported the same results. Dale and Brown [25] focused on the cost-effectiveness of the C&C Demonstration, but also conducted additional analyses to test the program’s effect on quality outcomes, including satisfaction with services. It was found that the treatment group reported being more satisfied with overall care than the control group. Differences were significant for the nonelderly subgroup in the three sites where C&C Demonstration was implemented.

Glendinning et al. [29] measured satisfaction with services and quality of services based on quality indicators derived from the National User Experience Survey. It was found that 49% of the individual budget group and 43% of the control group were either extremely or very satisfied with the help they received, and this was statistically significant. However, the result became non-significant when proxies were excluded. Differences between the treatment and control groups were not significant for the disability subgroups, except for young, physically disabled people in the individual budget group who were significantly more likely to report higher quality of care. Wiener et al. [34] reported on satisfaction amongst consumer and agency-directed care recipients by constructing an eight-item Satisfaction with Paid Personal Assistance scale (SPPAS). Outcomes for two age groups, younger and older than 65, were reported. For the younger cohort, average SPPAS ratings were higher for consumer-directed participants than agency-directed ones, but the difference was not significant.

Unmet needs:

Benjamin et al. [16] compared unmet needs for ADLs and IADLs between consumer- and agency-directed participants. Agency-directed participants reported significantly fewer ADL unmet needs than those receiving consumer-directed care. Consumer-directed participants reported fewer IADL needs going unmet compared with their agency-directed counterparts, but the difference was not significant. Brown et al. [17] reported the unmet needs of adults for help with daily living activities, household activities, transportation, and routine health care. Compared with the control group, significantly fewer people in the treatment group in all three states had unmet needs for help with daily living activities. Additionally, in the three states, treatment group participants were significantly less likely to report unmet needs for help with household activities or transportation. Adults in New Jersey and Florida were significantly less likely to report unmet needs with routine health care. However, whilst treatment group participants in Arkansas also reported lower unmet needs in this domain, the difference was not significant. However, treatment–control differences in unmet need related questions were mostly not significant when considering sample members with proxy respondents [19]. Dale and Brown [25] reported the unmet needs of adults for personal care. In all three sites, treatment group nonelderly participants were significantly less likely to report unmet needs in personal care.

Costs:

In the US, the Cash and Counseling demonstration estimated annual expenditure for both Medicaid and Medicare services [25]. Medicaid is an assistance program that serves low-income people. It is a federal program varying state by state and run by local authorities within federal guidelines. Usually, there is no out-of-pocket cost for medical expenses. Medicare is an insurance program providing coverage for people aged 65 and over, regardless of their income, and for young disabled individuals. Patients are required to pay a part of the costs as for inpatient hospital care. The total combined Medicaid and Medicare expenditure was significantly higher for the treatment group for adults in Florida, while no significant difference was found in Arkansas and New Jersey. Further results, broken down by service type, are provided in Appendix A. Costs per recipient per month were also reported for adults at each demonstration site. For all states, costs were significantly higher for the treatment group (see Appendix A,).

In the UK, the Individual Budget evaluation reported on costs for three cost categories: social care, health care, and care and support planning and management [29]. The difference in mean weekly social care costs between the treatment and control groups was small for all disability cohorts and not statistically significant for any, including the total sample (including older people) (see Appendix A). Average weekly costs, disaggregated by service type, are detailed in Appendix A.

For health care costs, the mean weekly health care cost for the individual budget group (£83) was significantly higher than for people in the control group (£59); however, these figures include the older people subgroup. When broken down by health resource, the only significant difference between the treatment and control groups was the cost of in-patient stays in hospitals, but again, this could be due to the older sub-population (see Appendix A).

Finally, analysis of the care and support planning and management costs found that the average weekly care management cost of the treatment group was significantly higher (£18) than the control group (£11). Glendinning et al. [29] also conducted a cost-effectiveness analysis using both the ASCOT and GHQ quality-of-life measurement scales, for which they found slightly different results. Using the ASCOT score measured across the total sample, including older people, individual budgets appeared to be marginally more cost-effective than the comparator (i.e., standard arranged support). When using the GHQ score, there was no significant difference in the cost-effectiveness between the two schemes. In terms of disability subgroups, it was found that the cost-effectiveness evidence supporting individual budgets was strongest for the mental health service users. There was limited evidence to support cost-effectiveness amongst people with physical or learning disabilities.

Woolham and Benton [35] reported mean weekly costs for both personal budget holders and traditional service users in three disability groups: learning disabilities, mental health conditions, and physical disabilities. Costs were higher for the personal budget holders in all disability groups, but no further statistical tests were conducted to check for significance (see Appendix A).

A cost-effectiveness analysis was also conducted using both ADL and GHQ scores. The analysis was performed on the total sample, including older people, and results were only presented in the form of scatterplots. The findings indicated a significant increase in the costs for the personal budget scheme compared to the traditional scheme, combined with a small benefit for the former when benefits were measured by GHQ scores, but no visible difference in the case of ADL scores. The authors found that compared to younger adults, older people did not greatly benefit from possessing a budget on the outcome measures used, but costs were higher for budget holders across all care groups.

Overall, while costs for personal assistance services and overall costs increase due to personal budgets, reduced costs are seen for nursing facility care and hospital inpatient stays. Two articles have investigated the cost-effectiveness of personalized budgets, with one providing results in favour of individual budget being a cost-effective solution for mental health service users.

### 3.4. Outcomes for Caregivers

The Cash and Counseling demonstration conducted computer-assisted telephone interviews with caregivers 10 months after random assignment, to estimate and compare program effects on outcomes for caregivers of treatment group members with those for the caregivers of control group members [36]. Outcomes for caregivers of nonelderly adults were only available for the Arkansas site. Sample sizes varied from measure to measure, because certain survey questions were asked only of sample members who met certain criteria and the largest sample consisted of 1433 members. Hours of care provided were calculated from the total hours of care provided in a two-week reference period for both live-in and visiting caregivers. Live-in caregivers were found to provide significantly fewer hours of care for those in the treatment group compared with the control group. Visiting caregivers also provided fewer hours of care, but the difference was not significant. The percentage of caregivers satisfied with the care recipient’s overall care was significantly higher in the treatment group than the control group. Caregiver well-being was calculated based on the three indicators of emotional strain, financial strain, and physical strain. Caregivers for the treatment group reported significantly lower levels of strain on all three indicators.

As part of the Individual Budgets pilot programme in England, structured interviews were conducted with 129 caregivers to determine the positive and negative effects that individual budgets could have on caregivers [37]. Caregiver quality of life was measured on a seven-item scale from ‘*so bad it could not be worse’* to ‘*so good it could not be better’*. Caregivers who provided assistance to the individual budget group were significantly more likely to report higher quality of life compared with this in the comparison group. Caregiver well-being was measured with the General Health Questionnaire and, although outcomes appeared better for the caregivers of the individual budget group, no statistical difference between the treatment and control groups was found.

Five domains of the Social Care Outcomes Toolkit (ASCOT) are relevant for caregivers: social participation and involvement, control over daily life, safety, occupation and employment, and caring role. Although caregivers of individual budget holders appeared to have better overall outcomes, no significant difference was found between the treatment and control groups. However, when broken down by the five ASCOT domains, caregivers in the treatment group were significantly more likely to report that they were fully occupied in activities of their choice. Impact on caregivers was measured using the Carers of Older People in Europe Scale (COPE Index), which measures impact on three components: negative impact on caregiving, positive aspects of caregiving, and quality of support. While no statistical significance was reached on any component, there was evidence that caregivers of individual budget holders were more likely to view their caregiving role positively compared with the control group. Finally, caregivers’ satisfaction with services was measured on a seven-point scale from ‘*very dissatisfied’* to ‘*very satisfied’*, and no statistically significant difference in satisfaction was found between the two groups.

## 4. Discussion

This review of the effectiveness and costs of personalized budgeting identified 23 quantitative articles that met or exceeded our minimum quality threshold. The studies reported outcomes for service users, caregivers, and service costs. Overall, the evidence base of quantitative studies, established by the search strategy outlined in this report, is limited in size and quality, although the evidence base has grown somewhat compared to previous reviews.

The present review adds to the previous review literature in several ways: first, it is the most up-to-date and most comprehensive one in scope. Harkes et al. [8] conducted a systematic review of self-directed support for people with learning disabilities and exclusively on UK studies, of which only two quantitative ones were identified. The review by Webber et al. [7] zoomed in on the effectiveness of personal budgets for people with mental health conditions. The scope in Fleming et al. [9] was comprehensive in terms of types of disability, but the primary studies reviewed reached only until 2016, identifying seven quantitative studies from the US and UK only.

While the additional years and broader scope that we have added has—not surprisingly—revealed more evidence, the overall amount of the existing evidence on the effectiveness—and even far more so, on the cost-effectiveness—of the personal budgeting base remains limited. This is a conclusion that is in line with previous reviews [7,8,9]. In contrast to the previous reviews, we also present and discuss our findings by disability type, to detect potentially relevant heterogeneities. As discussed above, this has revealed some interesting, potentially policy-relevant insights, especially with regard to a possibly more nuanced effectiveness of personal budgets for people with intellectual disabilities, and, hence, the need to consider suitable adaptations of the policy to this user group.

While more quantitative studies have been published on the topic than we ultimately included, several of those did not meet the inclusion criteria, mainly due to the absence of a comparator. While the studies reporting outcomes for mixed disability cohorts often present results by disability subgroup, caution must be exercised in drawing comparisons, as the group assignment was invariably performed—either through randomization or other methods—between people using personal budgets and those using traditional services, and not between different disability groups.

The evidence overwhelmingly suggested that personalized budgets have led to an improvement in the selected outcomes, measured mostly in terms of service satisfaction and well-being. Quality-of-life outcomes were frequently assessed, with ten articles reporting such measures. Significantly improved quality-of-life outcomes were reported for all subgroups, except those with a physical disability, and positive outcomes were especially encountered in the areas of service users’ control over decisions and daily life, self-reported quality of life, improved self-esteem, and psychological well-being. However, there was some evidence indicating reduced quality of life for people with mental health conditions in receipt of a personal health budget, in terms of increased risk for cognitive problems or problems related to hallucinations and delusions, in addition to a higher risk for problems with activities of daily living. These were the only negative findings stemming from a relatively small programme confined to a single city [26].

Service users also tend to be more satisfied with services as well as with the care provided, which are the most reported outcomes across the 23 studies, with 15 covering measures in this domain. All disability subgroups, in addition to the mixed disability studies, reported significantly positive outcomes for service users in terms of increased satisfaction with their care and/or with the personal assistant’s performance. Specific areas of greater satisfaction include satisfaction with the choice and control over care received, availability and schedule of assistant, and with help received for household tasks.

Service users also experienced fewer adverse events, with lower unmet needs for the intellectual and developmental disability and mixed disability subgroups. Budget recipients recorded fewer unmet needs for a range of services, including occupational therapy and social and recreational activities, in addition to needs for help with daily living activities, household activities, transportation, and routine health care. However, one article focusing on consumer-directed care for mixed disabilities found that treatment group participants recorded a greater number of ADL needs going unmet due to not having help.

Only two articles covered caregiver outcomes, pointing to an under-studied research domain. Both reported significant benefits in terms of quality of life as well as in terms of satisfaction with services for caregivers of people benefiting from a personalized care offer. They also provide fewer hours of care on average when cohabitating, and so have more freedom in managing their time.

While an overall increase in service user benefits seems apparent, the data on costs and service use are less conclusive, based on the limited work that has assessed cost or cost-effectiveness dimensions. For mental health service users, one article found significantly lower costs were achieved for treatment group participants using clinical outpatient services, and another showed significant reduction over time in indirect costs. In apparent contrast, another study found significantly increased service use for such services. This differing result could be an artefact of the large heterogeneity across program designs, as the latter aimed to increase referrals to community-based services for individuals with serious mental health conditions through increased engagement with outpatient services. Hence, increases in costs and service use appear to be heavily dependent on the stated goals and aims of the specific program under consideration.

For mixed disability groups, treatment group participants reported lower costs for home health services, nursing facilities, and inpatient care, while significantly increased costs were recorded for personal assistance and care management services. As for people with intellectual and developmental disabilities, benefitting from personalized solutions such as individual budgets increase their satisfaction with the quality of care but also raises associated costs. Overall, three articles conducted a cost-effectiveness analysis to examine whether increased user benefits from personal budgets could be achieved at an acceptable cost compared with alternative services. The cost-effectiveness evidence supporting personal budgets was strongest for mental health service users.

Strengths and methodological limitations:

This review follows a state-of-the-art search strategy using the PICOS framework. Since the research on personalized budgeting is fairly recent, the exact terminology used varies across studies. To overcome this issue, we defined a large set of search terms to cover as many studies as possible. Consequently, more than 2100 studies were screened at the level of title and abstract. Moreover, this review considers studies not only in English but also in French and German, hence, going beyond most previous systematic reviews that were limited to English-only studies. We have also reviewed grey literature, representing a large source of evidence in policy evaluations, to reduce concern over publication bias.

Despite our efforts at being comprehensive and multilingual, more than 70% of the articles focus on the US, thereby limiting the universal generalizability of the findings.

Any systematic review is also inevitably limited by the quality of the studies reviewed, and its findings should be interpreted in light of methodological shortcomings. For instance, only 7 articles, out of the 23 reviewed, performed a sample size calculation to ensure adequate statistical power to detect meaningful effects. Therefore, outcomes could have been underestimated amongst small samples by failing to achieve statistical significance or, in contrast, larger sample sizes could have led to an amplification of effects.

No studies attempted to blind those measuring outcomes at follow-up, leading to a potential source of bias. Other methodological shortcomings include a failure to anticipate a sufficient roll-out time for pilot studies, resulting in many participants in the treatment group not receiving their personal budget or care for part or all of the pilot. The ten RCTs in this evidence base dealt with this issue by adopting an intent-to-treat analysis, measuring the effect of having the opportunity to receive a personal budget. This may have led to an underestimation of effects.

There is also a lack of evidence in relation to longer-term outcomes, with studies varying in duration, from as little as one month to nine years, with an average of approximately two years. Furthermore, many studies, specifically those of a cross-sectional design, did not specify the duration of time that service users were in receipt of their personal budget or care. The transparency and potential impact of proxy respondents was also lacking in many studies, with several articles excluding results using proxy respondents, thereby neglecting the input of a particularly vulnerable group of PwDs who could not communicate independently.

Finally, there is an inherent selection limitation in that participant recruitment in social care interventions is oftentimes voluntary, resulting in participants who may be in favour of the intervention. This was the case for many interventions included here, even in those where randomization to treatment and control groups was conducted, which inevitably were working with a sample of individuals who indicated a willingness to try a personalized budget. Such a sample fails to consider persons not interested or eager to change their current care arrangement and who, for ethical reasons, are not obliged to participate in the trial.

## 5. Conclusions

While the direct comparison of the findings between articles is inevitably compromised by considerable heterogeneity in programme designs and contexts, they share the common feature of a human rights-inspired desire to give more choice and control, and hence, self-determination, to the service user, in accordance with the UNCRPD. Implementation models ranged across a broad spectrum of service delivery, from allowing the consumer to direct and manage their care, to allocating funds by means of a personalized care plan, to explicit cash allowances with variable levels of flexibility in spending.

Despite wide variations in terms of study type and quality, targeted population, and programme implementation processes, some general conclusions may be drawn. Most frequently, personalized budgets seem to offer both service users and caregivers better quality of life, through increased control and decision-making and greater satisfaction with care. Limited negative effects, such as a higher risk of cognitive problems, were reported only for people with mental disabilities in some cases, implying that the optimal design of consumer-directed care in this case might require further thought. While individual benefit outcomes have largely been covered, only a small number of studies reported cost outcomes, finding mixed results. Specifically, there is evidence for potential reduced costs for mental health clinical outpatient services, but also for significantly increased costs of other services when considering mixed disability subgroups. This may be due to the failure of the traditional health care system to provide adequate care to PwDs [25].

Some—though not all—of the very few studies examining cost-effectiveness do find personal budgets to be more cost-effective than alternative options, implying that their potentially higher costs may be more than outweighed by additional benefits. Some evidence also points to significant reductions in certain service use areas, which at least hints at the potential that personalized budgeting may entail not only better outcomes than alternative arrangements, but also—in certain cases—possibly reduced costs. The limited evidence base precludes, however, any generalization of the cost-effectiveness findings in the literature. In addition, the heterogeneity of the results on costs suggests a complex interaction between the cost of personalization and the user population. Research on a variety of models fitting to the needs of people with different expectations and disability types is warranted. A drive towards harmonizing outcome measures would also benefit the interpretation of findings in this research field and enable better comparison between interventions of different designs.

## Figures and Tables

**Figure 1 ijerph-19-16225-f001:**
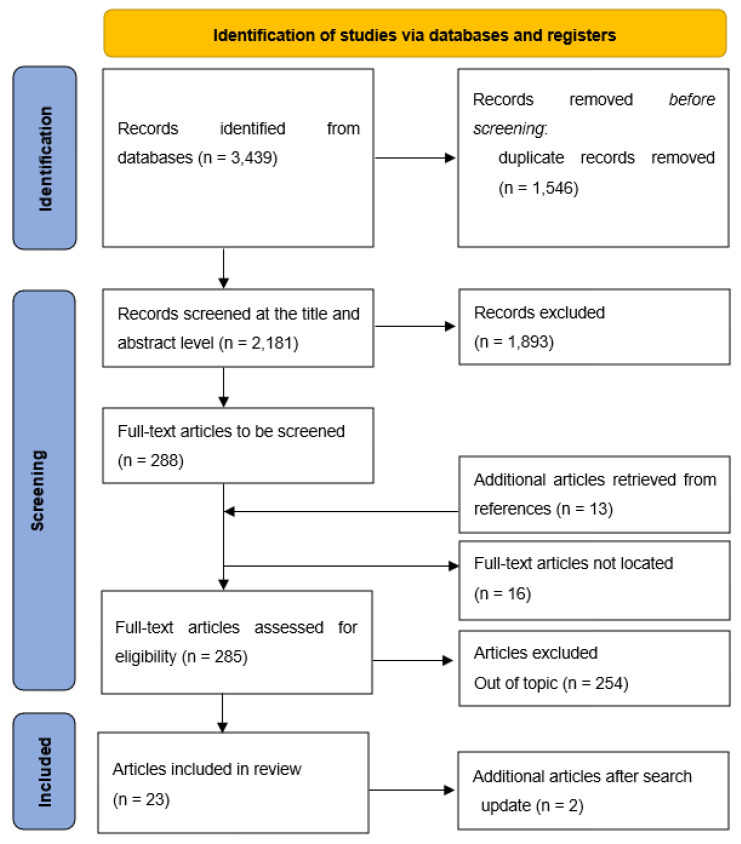
PRISMA flow diagram.

**Figure 2 ijerph-19-16225-f002:**
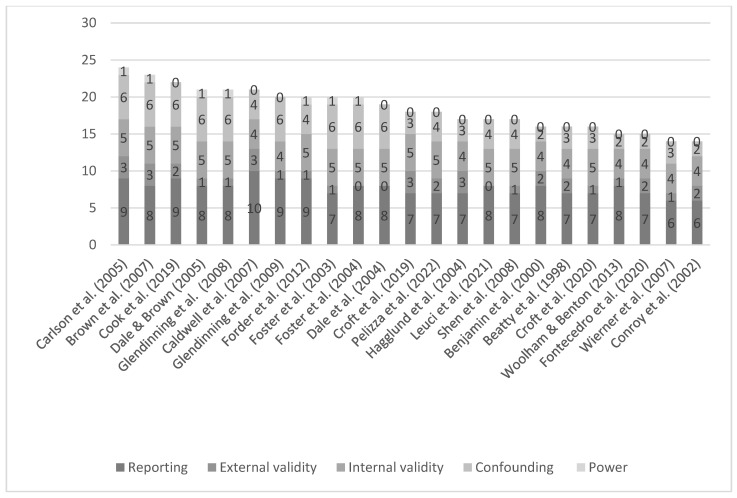
Quality assessment scores from the Downs and Black checklist.

**Table 1 ijerph-19-16225-t001:** Summary of inclusion and exclusion criteria for screening.

	Inclusion Criteria	Exclusion Criteria
**Language**	English, German, French	All other languages
**Location**	OECD high-income countries	
**Study type**	Quantitative, mixed methods	Qualitative
**Population**	Children and adolescents Adults younger than 65 years Elderly (typically over 65 years) with the onset of disability prior to 18 years of age	Elderly people (if more than 50% of the total study population)
**Types of disability**	Mental health, Autism, Disabilities of physical, intellectual, developmental, or sensory type	Age-related disabilities such as dementia
**Timeframe**	1985 onwards	Pre-1985
**Funding source**	State funded	Privately funded
**Comparator**	Presence of a control group or before–after comparative study design	No comparison

**Table 2 ijerph-19-16225-t002:** Article characteristics for service user and caregiver outcomes.

Author(Year)	Location; Study Design; Disabilities; Duration	Sample Size; % Female; Age (Mean)	Participant Groups	Outcomes of Interest Reported	Quality Score	Intervention
Beatty et al. (1998) [15]	Virginia, US; Quasi-experimental longitudinal comparative study; Physical; n/a cross-sectional study design	92, (T) 60, (C) 32; (T) 47%, (C) 42%; (T) 41.7, (C) 43.7	(T) Consumer-directed personal assistance service, in which users could hire, fire, and train their attendants. (C) On the waiting list for consumer-directed care.	Satisfaction with services.	Fair	Personal Assistance Services Program (PAS)
Benjamin et al. (2000) [16]	California, US; Controlled cross-sectional study; Mixed; n/a cross-sectional study	1095, (T) 511, (C) 584; (T) 69.9%, (C) 76.8%; Mean age not reported, (T) 53.6% ≥65 yrs, (C) 50% ≥65 yrs.	(T) Consumer-directed in-home supportive services (IHSS), where users could hire anyone they chose to provide care. (C) Receiving care under the professional home-care agency model.	Service satisfaction, unmet needs.	Fair	In-Home Supportive Services (IHSS)
Brown et al. (2007) [17]	Arkansas, New Jersey, Florida, US.; RCT; Mixed; 9 months;	2825: Arkansas (T) 243, (C) 230, New Jersey (T) 345, (C) 337, Florida adults (T) 419, (C) 392, Florida children (T) 441, (C) 418; Arkansas 67.6%, New Jersey 66.1%, Florida children 37%, Florida adults 45.4%; Mean age not reported, Arkansas 18–39 yrs 27.5%, 40–64 yrs 72.5%, New Jersey 18–39 yrs 33.8%, 40–64 yrs 66.2%, Florida children 3–12 yrs 71.2%, 13–17 yrs 28.8%, Florida adults 18–39 yrs 75.5%, 40–59 yrs 24.5%;	(T) Consumer-directed care where participants receive a monthly allowance to hire workers of their own choosing and to purchase care-related services and goods. (C) Received personal care services or home- and community-based services as usual.	Satisfaction with services, unmet needs, costs.	Good	Cash and Counseling (C&C)
Caldwell et al. (2007) [18]	Illinois, US; Longitudinal comparative before-after study; Developmental; 9 years	87, (T) 38, (C) 49; 49.4%; (T) 36.59, (C) 27.78	(T) Consumer-directed program where families were provided with an individualised budget and decided what services and supports to purchase. (C) Families on the waiting list for the program.	Service satisfaction, unmet needs.	Good	Home Based Support Services Program (HBSSP)
Carlson et al. (2005) [19]	Arkansas, New Jersey, Florida, US; RCT; Mixed; 9 months;	1966: Arkansas (T) 243, (C) 230, New Jersey (T) 345, (C) 337, Florida adults (T) 419, (C) 392; Arkansas 67.6%, New Jersey 65.1%, Florida adults 45.4%; Mean age not reported, Arkansas 18–39 yrs 27.1%, 40–64 yrs 72.9%, New Jersey 18–39 yrs 34.9%, 40–64 yrs 65.1%, Florida adults 18–39 yrs 75%, 40–59 yrs 25%;	(T) Consumer-directed care where participants receive a monthly allowance to hire workers of their own choosing and to purchase care related services and goods. (C) Received personal care services or home- and community-based services as usual.	Service satisfaction, quality-of-life, unmet needs.	Excellent	Cash and Counseling (C&C)
Conroy et al. (2002) [20]	California, US; Controlled before-after study; Intellectual disabilities; 2 years	77, (T) 63, (C) 14; (T) 28.6%, (C) 35.7%; (T) 25.4, (C) 27.9	(T) Self-determination program in which participants received an individual budget. (C) A group of people who wanted to participate, receiving traditional services	Choice, perception of quality-of-life, adaptive and challenging behaviours.	Fair	Pilot project on self-determination
Cook et al. (2019) [21]	Texas, US; Randomised controlled trial; Mental illness; 24 months	216, (T) 114, (C) 102; 62%; 41.6 yrs	(T) Received self-directed care by means of an individual budget for the purchase of services and goods corresponding to plan goals. (C) Services as usual	Self-perceived recovery, psychosocial status, psychiatric and somatic symptoms, satisfaction with services, costs of services.	Good	Self-Directed Care Program (SDC)
Croft et al. (2019) [22]	Pennsylvania, US; Uncontrolled pre-post study; Mental health conditions; 3 years	45; 71.1%; 51.5 yrs	(T) Self-directed care in which participants banked funds by intentionally reducing their use of some mental health services and applying the cost savings towards flexible spending of approved nonclinical goods and services.	Service use, costs.	Fair	Consumer Recovery Investment Fund Self-Directed Care II (CRIF-SDF II)
Croft et al. (2020) [23]	Utah, US; Quasi-experimental before-after comparative study; Mental health conditions; Enrolment average of 199 days	623, (T) 94, (C) 529; (T) 38.3%, (C) 38.37%; (T) 42.38, (C) 42.85	(T) Self-direction in which funds were allocated from a flexible budget to meet recovery goals. (C) Received traditional Medicaid-funded and state-funded services.	Service use.	Fair	Mental Health Access to Recovery (MHATR)
Dale et al. (2004) [24]	Florida, US; RCT; Children, Developmental; 9 months	1002, (T) 501, (C) 501; (T) 38.1%, (C) 35.9%; <12 yrs (T) 63.5%, (C) 64.1%;	(T) Parents of treatment group members were given the opportunity to receive a monthly allowance they could use to hire their choice of caregivers or to buy other services or goods to meet their child’s care needs. (C) Received traditional waiver services.	Costs.	Good	Cash and Counseling (C&C)
Dale & Brown (2005) [25]	Arkansas, New Jersey, Florida, US; RCT; Mixed; 1 to 2 years;	2282: Arkansas 556, New Jersey 813, Florida adults 913; Arkansas 67.6%, New Jersey 66.1%, Florida adults 45.3%; Mean age not reported, Arkansas 18–39 yrs 27.5%, 40–64 yrs 72.5%, New Jersey 18–39 yrs 33.7%, 40–64 yrs 66.31%, Florida adults 18–39 yrs 75.5%, 40–59 yrs 24.5%;	(T) Consumer-directed care where participants receive a monthly allowance to hire workers of their own choosing and to purchase care related services and goods. (C) Received personal care services or home- and community-based services as usual.	Costs, satisfaction with care, unmet needs.	Good	Cash and Counseling (C&C)
Fontecedro et al. (2020) [26]	Italy; Observational comparative cross- sectional study; Mental health conditions; n/a cross-sectional study design	128, (T) 67, (C) 61; (T) 37.3%, (C) 45.9%; Mean age not reported. 20–59 yrs (T) 74.6%, (C) 65.4%	(T) Received an individual budget. (C) Care maintained as usual.	Health of the Nation Outcome Scale	Fair	Individual Health Budget (IHB)
Forder et al. (2012) [27]	England; RCT; Mental health conditions; 12 months	197, (T) 105, (C) 92; (T) 49%, (C) 50%; (T) 45 yrs, 11% ≥75 yrs, (C) 53 yrs, 10% ≥75 yrs	(T) Received a personal health budget. (C) Continued conventional support arrangement.	Care-related quality-of-life, health-related quality-of-life, psychological well-being, subjective well-being, costs, cost-effectiveness.	Good	Personal HealthBudget Pilot Programme (PHB)
Foster et al. (2004) [28]	Florida, US; RCT; Children, Developmental; 9 months	859, (T) 441, (C) 418; (T) 38.5%, (C) 35.2%; < 12 yrs (T) 63.3%, (C) 63.4%;	(T) Consumer-directed care where parents could use the allowance to hire their choice of caregivers and to buy other services and goods to meet their child’s care needs. (C) Received personal care services or home- and community-based services as usual.	Satisfaction with child’s care, child’s unmet needs, child’s quality of life.	Good	Cash and Counseling (C&C)
Glendinning et al. (2008) [29]	UK; Randomised controlled trial; Mixed; 6 months	959, (T) 510, (C) 449; 56%; 57 yrs	(T) Participants received an individual budget in addition to traditional social care services. (C) Continued to receive traditional social care support.	Perceived quality of life, psychological well-being, social care outcomes, self-perceived health, satisfaction with services, costs, cost-effectiveness.	Good	Individual Budgets Pilot Programme (IBPP)
Hagglund et al. (2004) [30]	Missouri, US; Controlled cross-sectional study; Physical; Enrolled for a minimum of 1 month	114, (T) 61, (C) 53; 32%; 48 yrs	(T) Consumer-directed personal assistance services in which consumers hired and managed their own personal assistants. (C) Received services through an agency-directed model.	Unmet needs, satisfaction, quality-of-life	Fair	Assistance Services programme (PAS)
Leuci et al. (2021) [31]	Parma, Italy; Longitudinal comparative before-after study; Mental health conditions; 2 years	104, (T) 49, (C) 55; 36%, (T) 26.5%; (C) 43.6; 28 yrs, (T) 26 yrs, (C) 31 yrs.	(T) Received a personal health budget as a part of a specific program. (C) Care maintained in the specific program without a personal budget.	Brief Psychiatric Rating Scale, Global Assessment of Functioning, Health of the Nation Outcome Scale	Fair	Personal Health Budget (PHB)
Pelizza et al. (2022) [32]	Parma, Italy; Uncontrolled pre-post study; Mental health conditions; 2 years	137; 38%; 33 yrs.	(T) Received a multi-axis personal health budget.	Brief Psychiatric Rating Scale, Global Assessment of Functioning, Health of the Nation Outcome Scale	Fair	Personal Health Budget (PHB)
Shen et al. (2008) [33]	New Jersey, US; RCT; Mental health conditions; 9 months	228, (T) 109, (C) 119; (T) 77%, (C) 64%; (T) 18–39 yrs 31%, 40–64 yrs 69%, (C) 18–39 yrs 29%, 40–64 yrs 71%.	(T) Consumer-directed care where participants receive a monthly allowance to hire workers of their own choosing and to purchase care related services and goods. (C) Received personal care services or home- and community-based services as usual.	Satisfaction with services, satisfaction with quality-of-life.	Fair	Cash and Counseling (C&C)
Wiener et al. (2007) [34]	Washington State, US; Controlled cross-sectional study; Various; n/a cross-sectional study design	229, (T) 124, (C) 105; 62.9%; Mean age not reported. 28% ≤44 yrs, 72% aged 45–64.	(T) Consumer-directed personal assistance services where consumers are responsible for hiring, orienting, supervising, and finding replaces for the caregiver. (C) Receiving agency-directed care.	Satisfaction with services.	Fair	Home- and Community-Based Services (HCBS)
Woolham and Benton (2013) [35]	England; Controlled cross-sectional study; Various; n/a cross-sectional study design	402, (T) 126, (C) 276 (under 65 yrs); (T) 66.1%, (C) 35.8%, (includes over 65 yrs); (T) 51.5 yrs, (C) 54.9 yrs (includes over 65 yrs)	(T) Received a personal budget. (C) Received traditional services.	Psychological well-being, activities of daily living, costs, cost-effectiveness.	Fair	Personal Budget (PB)
Foster et al. (2003) [36]	Arkansas, US;	Survey of 1433 carers; 39 or younger 22.5%, 40–64 yrs 64.1%, 65 or older 13.5%.	(T) Consumer-directed care where parents could use the allowance to hire their choice of caregivers and to buy other services and goods to meet their child’s care needs. (C) Received personal care services or home- and community-based services as usual.	Hours of care provided, satisfaction, well-being.	Good	Cash and Counseling (C&C)
Glendinning et al. (2009) [37]	UK;	Carers: 129, (T) 69, (C) 60, Physical disabilities (T) 8, (C) 11, Older people (T) 16, (C) 17, Learning disabilities (T) 32, (C) 38, Mental health conditions (T) 4, (C) 3 All ≥25 yrs, 45–59 yrs, (T) 57%, (C) 58%, ≥60 yrs, (T) 32%, (C) 36%.	(T) Participants received an individual budget in addition to traditional social care services. (C) Continued to receive traditional social care support	Quality-of-life, well-being, social care outcomes, self-perceived health, impact, satisfaction with services.	Good	Individual Budget Pilot Programme (IBPP)

Note: T = treatment group; C = control group.

**Table 3 ijerph-19-16225-t003:** Percentage of elderly participants reported in three studies providing this detail.

Article	Age Group	Treatment (%)	Control (%)
Benjamin et al. (2000) [16]	≥65 years	53.6	50
Fontecedro et al. (2020) [26]	≥60 years	25.4	34.4
Forder et al. (2012) [27]	≥75 years	11	10

**Table 4 ijerph-19-16225-t004:** Benefits and disadvantages associated with personal budgets.

Outcome	Article	Benefits Associated with Personal Budgets	Disadvantages Reported with Personal Budgets
		**Service Users’ Outcomes**	
*Physical disabilities*			
Service satisfaction	Beatty et al. (1998) [15] Hagglund et al. (2004) [30]	More satisfied with the costs of care, choice, and control over care, control over assistant’s work schedule, authority to direct assistant, and the availability of assistant in both everyday care and emergency situations. Greater satisfaction with daily living and community living.	
*Intellectual and developmental disabilities*			
Service satisfaction	Caldwell et al. (2007) [18] Foster et al. (2004) [28]	Increased satisfaction with services over time. Increased satisfaction with caregiver’s schedule, relationship with the caregiver, help around the house and community, and overall care arrangements.	
Unmet needs	Caldwell et al. (2007) [18] Foster et al. (2004) [28]	Fewer service needs for occupational therapy, social/recreational activities, educational/vocational training, assistance obtaining benefits, and assistance obtaining vocational services. Fewer unmet needs for help with daily living activities, household activities, transportation, and routine health care.	
Quality of life	Conroy et al. (2002) [20] Foster et al. (2004) [28]	Increased control and power over decisions. Better perception of quality of life. Some evidence of increases in adaptive behaviour. Higher child’s quality of life.	
Cost	Dale et al. (2004) [24]		Higher monthly costs and annual expenditures.
*Mental health disabilities*			
Quality of life	Cook et al. (2019) [21] Fontecedro et al. (2020) [26] Leuci et al. (2021) [31] Pelizza et al. (2022) [32] Shen et al. (2008) [33]	Higher perceived level of recovery, especially in the domains of goal orientation and personal confidence. Improvements in self-esteem, coping mastery, and perception of service delivery being supportive of personal autonomy. Lower somatic symptom severity. Less likely to be at risk for problems in the availability of resources for work and recreation activities. Improvements in global functioning and in the domains of impairment, psychiatric symptoms, and behavioural problems. Improvements in negative symptoms and social behaviour. Higher level of satisfaction with life.	Increased risk of severe to very severe cognitive problems and problems related to hallucinations and delusions. Higher risk for moderately severe problems with activities of daily living.
Service satisfaction	Cook et al. (2019) [21] Shen et al. (2008) [33]	Higher satisfaction with mental health services. Higher satisfaction with their caregiver’s schedule, help around the house and community, and overall care arrangements.	
Service use	Croft et al. (2020) [23]		Increased use of rehabilitation and outpatient treatment services.
Cost	Croft et al. (2019) [22] Forder et al. (2012) [27]	Reduced costs for mental health clinical outpatient services. Significant reduction over time in indirect costs. Cost-effective when using ASCOT scale.	
*Mixed disabilities*			
Quality of life	Glendinning et al. (2008) [29] Woolham and Benton (2013) [35]	Higher self-reported quality of life for mental health service users. More likely to feel in control of daily life. Better psychological well-being.	
Service satisfaction	Benjamin et al. (2000) [16] Brown et al. (2007) [17]; Carlson et al. (2005) [19] Dale and Brown (2005) [25] Glendinning et al. (2008) [29]	More positive ratings for technical quality of services, service impact, general satisfaction, and caregiver’s interpersonal manner. More satisfied with caregiver’s schedule, relationship with caregiver, help with Daily Living Activities, help around the house/community, overall care arrangements, and help with routine health care. Higher satisfaction with overall care. Higher satisfaction with quality of care for young physically disabled adults.	
Unmet needs	Benjamin et al. (2000) [16] Brown et al. (2007) [17] Dale and Brown (2005) [25]	Fewer unmet needs for help with daily living activities, household activities, transportation, and routine health care. Fewer unmet needs.	More ADL needs unmet.
Cost	Brown et al. (2007) [17] Dale and Brown (2005) [25] Glendinning et al. (2008) [29]	Lower costs for home health services, nursing facilities, and inpatient care. Lower costs for home care and Independent Living Fund.Some evidence of cost-effectiveness.	Higher monthly costs. Higher costs for personal assistance services and total care. Higher cost for personal assistants. Higher care management costs.
		**Caregivers’ outcomes**	
Quality of life	Foster et al. (2003) [36] Glendinning et al. (2009) [37]	Lower levels of emotional, financial, and physical strain.Higher quality of life. More likely to be fully occupied in activities of their choice.	
Satisfaction with services	Foster et al. (2003) [36]	More satisfied with service user’s overall care.	
Hours of care provided	Foster et al. (2003) [36]	Fewer hours of care provided by live-in caregivers.	

## Data Availability

Not applicable.

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
