# Peer review of "The Effects and Costs of Personalized Budgets for People with Disabilities: A Systematic Review"

_ijerph, 2022, doi:10.3390/ijerph192316225_

Round 1

Reviewer 1 Report

This review evaluated the effects of personalized budgets for people with disabilities (PwDs), in terms of a range of benefit and cost outcomes. Benefit metrics of interest comprised measures of well-being, service satisfaction and use, quality of life, health, and unmet needs, which is an interesting topic to explore. Below are the comments to improve the contents of the article.

1. The title of this article isEvaluating the effects and costs….” But how the authors evaluated the effects and costs was unclear. The authors reviewed and summarized the studies, not evaluated them. I suggest changing the title to a more accurate title.

2. In the methods section, the study design was missing. I recommend extracting some information in the last paragraph of the introduction section to include in the study design subsection.

3. It was unclear whether the authors used a qualitative or quantitative approach for this systematic review. The authors mentioned that “we add to previous work by taking into account also the most recent quantitative evidence that has attempted to evaluate outcomes, costs …..“ but I cannot find any quantitative result in the results section.

4. The authors mentioned heterogeneity in the results section but did not explain how heterogeneity was determined or measured in the methods section.

5. The authors also did not clearly explain the outcomes (what are the outcomes? How was the outcome determined/ measured?) in the methods section.

6. The results arrangement was not tally with the outcomes mentioned in the methods section. The explanation of the outcomes was not clear. The authors should explain clearly the outcomes. For example, the outcomes were divided into outcomes for service users and outcomes for caregivers (which was not mentioned in the methods section). From there, the outcomes for service users were divided into physical disability, intellectual and developmental disability, mental health condition, and mixed disability (also not mentioned in the methods section)

7. The author should include a comparison between this review with the other reviews or studies in the discussion section.

8. Table 5 to Table 7 and all appendices were missing in this article.

Reviewer 2 Report

General comment: First I would like to thank the editors and the authors for the opportunity to review this interesting manuscript. This is a well-written systematic review with an interesting research topic and suits to the topic of the special issue. However, I think there are two major concerns before publication: 1) the search needs to be updated, because from the period between 2020 to 2022 potential reports could be lost. 2) Discussion section has to be rewritten discussing the findings of the review with previous literature and also include a research implication section. Hope my comments help authors to improve the quality of this interesting manuscript.

Introduction:

# Comment 1: Line 72. Please, remove the point after the cite of Webber et al. 2014.

# Comment 2: Line 82. What did authors mean with the term rapid? Please, clarify.

# Comment 3: Line 82-101: To me it seems that the ideas stated in this paragraph would be more adequate to the discussion section than in the current location. I do not know if it is the way it was written, but I think all these ideas suit better in discussion or even in strengths of the review. I advise the authors to justify the need of the review with clear points and establish the study goals to help readers to understand the results of the review. Thanks

Methods:

# Comment 1: Was this systematic review registered in PROSPERO or in another registry?

# Comment 2: Line 135-142. Please, remove the commas after the points.

# Comment 3: One of the major issues of your review is that potential records could have been lost during the period of 2020 to 2022, so search needs to be updated.

Results:

# Comment 1: Title of the sub-section 3.3.2 is colored in blue not in black.

# Comment 2: Please, consider changing your current PRISMA flowchart to the new version of 2020

# Comment 3: congrats authors for Table 4, it is really clarifying.

Discussion:

# Comment 1: This is an unusual way to present a discussion. In its current form the discussion summaries the findings of the review, so information is duplicated. This section needs to be rewritten apporting discussion of the review findings with other literature and set hypotheses. Furthermore, a research implication sub-section would be really interesting for readers and I suggest authors add it within the discussion section. I think this suggestion could improve the quality of the manuscript.

# Comment 2: I congratulate the authors for the strength and limitation section. 

Round 2

Reviewer 2 Report

Dear authors,

I would like to thank you the enormous effort to address all my comments. I think the manuscript have improved. Nonetheless, before publication it is necessary to review some minors changes. Let my comments below:

Comment 1: It is necessary to improve Figure 1. It seems that some information is missing and need to be clarified. Please, also turn the title of Figure 1 into a caption.

Comment 2: Figure 2 title turn into a caption

Comment 3: What are the major findings of the Downs and Black Checklist? Please include this information in the results section.

Author Response

We appreciate the reviewer’s comments. Below we respond (in bold) to each point raised.

“Dear authors,

I would like to thank you the enormous effort to address all my comments. I think the manuscript have improved. Nonetheless, before publication it is necessary to review some minors changes. Let my comments below:”

Thanks for the positive overall feedback!

“Comment 1: It is necessary to improve Figure 1. It seems that some information is missing and need to be clarified. Please, also turn the title of Figure 1 into a caption.”

Apologies – something must have gone wrong with the formatting or lay-out, as the information was not really missing, but some elements had been "hidden". Some of the boxes in this Figure became smaller, and the numbers on the second line did not appear anymore. We have now tried to make sure that the figure appears on a single page and that all the numbers are visible. We have also turned the title into a caption.

“Comment 2: Figure 2 title turn into a caption”

We have now turned the title into a caption.

“Comment 3: What are the major findings of the Downs and Black Checklist? Please include this information in the results section.”

In our submitted manuscript, the major findings were relegated to the (rather long) Appendix, which may be why the reviewer had not seen it. We have now moved a version of the Appendix text into the main text. The text reads as follows:

The Reporting domain of the Downs and Black checklist assesses whether information provided in each article is sufficient to allow for an unbiased assessment of the findings. The maximum score obtainable is 11, and the average score of the 23 included articles was 7.74 with a range of 6-10. The main weaknesses in this domain were a lack of clearly defined confounders and the reporting of individuals that were lost to follow-up. All articles scored full marks for providing a comprehensive statement of their aim and clearly describing participant characteristics, main outcomes to be measured, intervention of interest and main findings.

The representativeness of the article findings and their generalizability to the target population is assessed in the External Validity domain. A maximum score of 3 is possible and the average scored was 1.56 with a range of 0-3. Five articles received the maximum score, with most others failing to report response rates.

Biases in the measurement of the intervention and outcomes are addressed in the Internal Validity domain of the checklist, with a maximum possible score of 7. For the implementation of personal budgets, blinding of the participants is not possible and all articles were graded 0 on this criterion. The average score (marked out of 7) was 4.61, with a range of 4-5. No article focused on interventions that attempted to blind those measuring the outcomes. Another limitation was the lack of adjustment for different lengths of follow-up of the participants, to account for cases where individuals were in receipt of the treatment for different lengths of time. All articles reported using validated and reliable outcome measures.

The Confounding domain measures bias in the selection of participants. Nine articles received the maximum score of 6, and the average score was 4.22, with a range of 2-6. The weakest point was the lack of concealing intervention assignments from both patients and health care staff until the recruitment was complete. Furthermore, only ten of the 23 articles randomly assigned individuals to treatment or control groups. The final domain of Statistical Power determines if a sufficient sample size to detect a behavioral important effect is considered. Only seven articles conducted such an analysis.